# Systemic COVID-19 Vaccination Enhances the Humoral Immune Response after SARS-CoV-2 Infection: A Population Study from a Hospital in Poland Criteria for COVID-19 Reimmunization Are Needed

**DOI:** 10.3390/vaccines10020334

**Published:** 2022-02-19

**Authors:** Piotr Kosiorek, Dorota Elżbieta Kazberuk, Anna Hryniewicz, Robert Milewski, Samuel Stróż, Anna Stasiak-Barmuta

**Affiliations:** 1Department of Emergency, Maria Sklodowska-Curie Bialystok Oncology Centre, Ogrodowa 12, 15-027 Białystok, Poland; 2Department of Clinical Immunology, Medical University of Białystok, Jana Kilińskiego 1, 15-089 Białystok, Poland; samuel.stroz@umb.edu.pl (S.S.); a.barmuta@umb.edu.pl (A.S.-B.); 3Department of Radiotherapy, Maria Sklodowska-Curie Bialystok Oncology Centre, Ogrodowa 12, 15-027 Białystok, Poland; dkazberuk@onkologia.bialystok.pl; 4Department of Rehabilitation, Medical University of Białystok, Jana Kilińskiego 1, 15-089 Białystok, Poland; anna.hryniewicz@umb.edu.pl; 5Department of Statistics and Medical Informatics, Medical University of Białystok, Jana Kilińskiego 1, 15-089 Białystok, Poland; robert.milewski@umb.edu.pl

**Keywords:** COVID-19, BNT162b2, vaccination, SARS-CoV-2, reimmunization

## Abstract

Systemic vaccination with the BNT162b2 mRNA vaccine stimulates the humoral response. Our study aimed to compare the intensity of the humoral immune response, measured by SARS-CoV-2 IgG, SARS-CoV-2 IgM, and S-RBD-neutralizing IgG antibody levels after COVID-19 vaccination versus after SARS-CoV-2 infection. We analyzed 1060 people in the following groups: convalescents; healthy unvaccinated individuals; individuals vaccinated with Comirnaty, AstraZeneca, Moderna, or Johnson & Johnson; and vaccinated SARS-CoV-2 convalescents. The concentrations of SARS-CoV-2 IgG, SARS-CoV-2 IgM, and S-RBD-neutralizing antibodies were estimated in an oncology hospital laboratory by chemiluminescent immunoassay (CLIA; MAGLUMI). Results: (1) We observed a rise in antibody response in both the SARS-CoV-2 convalescent and COVID-19-vaccinated groups. (2) The levels of all antibody concentrations in vaccinated COVID-19 convalescents were significantly higher. (3) We differentiated asymptomatic SARS-CoV-2 convalescents from the control group. Our analysis suggests that monitoring SARS-CoV-2 IgG antibody concentrations is essential as an indicator of asymptomatic COVID-19 and as a measure of the effectiveness of the humoral response in convalescents and vaccinated people. Considering the time-limited effects of post-SARS-CoV-2 infection recovery or vaccination and the physiological half-life, among other factors, we suggest monitoring IgG antibody levels as a criterion for future vaccination.

## 1. Introduction

Vaccination is the best means of preventing infectious diseases. Currently, the Pfizer-BioNTech vaccine BNT162b2 (Comirnaty) [1] is recognized as among the most effective vaccines in preventing SARS-CoV-2 infection and severe COVID-19 [2,3,4]. Rising levels of specific antibodies—IgM and IgG—and S-RBD-neutralizing IgG as a humoral immune response are effects of vaccination. A similar effect is observed after SARS-CoV-2 exposure (coronavirus disease 2019, COVID-19) [5]. It has been proven that in symptomatic SARS-CoV-2 infections, the level of S-RBD-neutralizing antibodies correlates with the severity of COVID-19 and with hospitalization [5,6]. This correlation was not observed either in asymptomatic convalescent COVID-19 patients or in healthy vaccinated individuals after the first dose of a systemic double-vaccination scheme [7,8]. Previous research provided evidence that after a second systemic dose of the Pfizer-BioNTech mRNA vaccine, neutralizing antibody levels were lower than those in vaccinated convalescent COVID-19 patients [8]. Higher levels of neutralizing antibodies after SARS-CoV-2 exposure and COVID-19 vaccination have been detected in individuals with autoimmune disease [9,10]. Seroconversion to neutralizing antibodies is gradual and depends on the patient’s clinical state resulting from immune defense mechanisms, such as neutralization, complement activation, and cell cytotoxicity (antibody-dependent cellular cytotoxicity, ADCC) [7,11,12]. In rare situations, SARS-CoV-2 infection gives rise to systemic inflammatory response syndrome (SIRS), with systemic inflammatory multiorgan dysfunction (MODS) [13]. Depending on the phase and intensity of the inflammatory response, administering convalescent serum plasma antibodies as a COVID-19 treatment may reduce or strengthen the inflammation [14]. There is much controversy involving the use of convalescent-derived serum plasma for COVID-19 therapy [14]. Serum-plasma antibodies are very effective in the early stage of COVID-19, when the inflammation process has not yet involved elements of the humoral response [12,14,15].

The evaluation of systemic vaccinations against SARS-CoV-2 invariably raises questions about seroprotective antibody concentration levels and their half-life [11]. Aging of the immune system is, on the one hand, a risk factor for contracting COVID-19 and, on the other hand, determines poor postvaccination response [16]. The question then arises as to whether elderly individuals require an additional dose of the vaccine [16,17]. Should the same question be asked about immunosuppressed patients, people with immunodeficiency syndromes, post-transplantation patients, or oncologic or dialysis patients? Many published papers have proven the safety of BNT162b2 vaccination in patients on dialysis [18] and after lymphoma therapy [19], as well as the safety of a third dose in kidney transplant recipients [20]. In healthy individuals, partial seroprotection, with a mean of approximately 53% (32–68%, confidence interval 95%), is reached 14 days after the first dose of the BNT162b2 vaccine. Seven days after the second dose, seroprotection reaches 95% [16]. Seroprotection is even higher for SARS-CoV-2 variants B.1.1.7 and B.1.351, with values of 75% and 97% after the first and second doses, respectively [21].

## 2. Materials and Methods

Our study observed a group of 1063 people (786 females (74%) and 277 males (26%)) in the age bracket between 18 and 89 years old. People over 50 years old constituted 45% (*N* = 479) of the group. Because of different postvaccination reactions, three subjects were considered early hyper-responders based on their IgM and IgG levels, and we excluded them from further evaluation. The final group of 1060 subjects (Figure 1, Table 1) was divided into five subgroups (G0–G5): The control group, G0 (*N* = 154), included unvaccinated persons who showed no clinical signs of COVID-19 and tested negative for SARS-CoV-2 by RT-PCR. Group G1 (*N* = 76) included fully symptomatic COVID-19 patients who tested positive for SARS-CoV-2 by RT-PCR. Group G2 included 472 healthy individuals vaccinated with both doses of Comirnaty. Group G3 included 42 persons vaccinated with AstraZeneca (*N* = 21), Moderna (*N* = 19), or Johnson & Johnson (*N* = 2) according to the vaccination scheme. Group G4 included 312 COVID-19 convalescents vaccinated with Comirnaty. Group G5 consisted of 4 COVID-19 convalescents infected with SARS-CoV-2 after a complete systemic vaccination cycle. Two positive RT-PCR tests confirmed infection.

After a retrospective analysis of concentrations of SARS-CoV-2 IgG antibodies in group G0 (*N* = 154), we identified subgroup G01, which comprised 43 persons who had concentrations of specific IgG antibodies that were higher than the cutoff value (0.2 AU/mL). The existence of specific SARS-CoV-2 antibodies was regarded as evidence of viral contact, and the patients were classified as SARS-CoV-2 convalescents with no symptomatic COVID-19 history. Finally, G0 = 111. Among 43 analyzed cases (G01), 13 persons were in the seroconversion phase (positive for both IgM > 1.0 AU/mL and IgG > 0.2 AU/mL), 30 persons were in a late phase of producing secondary antibodies (positive for IgG, negative for IgM), and 8 persons were in an early phase of the humoral viral response (positive for IgM, negative for IgG).

### 2.1. Materials

Our study materials were blood specimens collected through venipuncture sampling. The concentration of antibodies was evaluated 4 h after blood collection. If an immediate assessment was not possible, the serum was collected and stored at −80 °C.

### 2.2. Methods

All 1063 participants had elevated levels of IgM and IgG antibodies oriented specifically towards SARS-CoV-2; in 546 subjects, anti-S (S-RBD) IgG antibodies were detected by chemiluminescent immunoassay (CLIA; MAGLUMI, Snibe Diagnostic, Shenzhen, China).

Results greater than or equal to 1.0 AU/mL SARS-CoV-2 IgG, IgM, and S-RBD were considered indicative of a reaction and regarded as positive, according to the manufacturer’s protocol.

Of the study participants, 827 were vaccinated, including 787 persons who received Comirnaty. BNT162b2 is 95% effective in preventing COVID-19, and similar vaccine efficacy is observed across subgroups defined by age, sex, race, and ethnicity [1].

The Bioethics Commission of Medical University approved our research.

## 3. Statistical Analysis

Nonparametric statistical methods were applied for the case–control analyses because many of the considered variables were not normally distributed. The Mann–Whitney test was adopted to compare two groups to each other, whereas the Kruskal–Wallis test was applied to assess more than two groups. Results were considered statistically significant when *p* < 0.05. We used PQStat Software 2021 (PQStat Software, Poznan, Poland) and Tibco Statistics 13.3 (TIBCO Software Inc., Palo Alto, CA, USA) for statistical analyses.

## 4. Results

After evaluating the concentrations of specific antibodies in the control group of healthy unvaccinated subjects, we identified a subgroup of SARS-CoV-2 convalescents (G01) after an asymptomatic case of COVID-19. The concentration of IgG antibodies in that subgroup was comparable to the concentration observed in COVID-19 convalescents with fully symptomatic disease confirmed by a positive RT-PCR test.

Our research showed the presence of SARS-CoV-2 antibodies in both the convalescent and vaccinated COVID-19 groups, and the concentration of antibodies decreased with time in both groups (Figure 2).

Significantly higher IgG- and IgM-specific antibody concentrations were detected in vaccinated COVID-19 convalescents (G4) (Figure 3).

After we analyzed the strength of specific IgG and IgM generation after vaccination, we did not find any significant difference between the studied vaccines (Pfizer, AstraZeneca, and Moderna). The sample size of the Johnson & Johnson vaccine-recipient group (*N* = 2) was too small to render a meaningful result; hence, we excluded these individuals from further analysis.

Figure 4 shows the levels of antibodies in individuals evaluated at time intervals (a) after full vaccination and (b) after a positive RT-PCR test result for SARS-CoV-2 (COVID-19). The completed analysis shows a linear dependence over time for both vaccination response and natural response to viral infection after clinically mild or moderate COVID-19. We observed the effect mentioned above in each studied group for up to 180 days postexposure and up to 90 days postvaccination.

## 5. Discussion

Evaluating the concentrations of IgG and IgM antibodies specific to SARS-CoV-2 allowed us to identify a new subgroup among healthy unvaccinated controls: subjects who had a previous asymptomatic SARS-CoV-2 infection. Every patient in this subgroup presented lower concentrations of specific SARS-CoV-2 IgG antibodies than the cutoff established for a positive result in laboratory tests, which is between 0.2 and 1.0 AU/mL.

Based on the concentration of neutralizing antibodies, we propose the following hierarchy of activation of the humoral response: unvaccinated (G0) < convalescents (G1) < vaccinated (G2, G3) < vaccinated convalescents (G4) < patients with SARS-CoV-2 infection after being fully vaccinated (G5).

Group G4 was exposed to the virus three times: the first time was when they were infected with the SARS-CoV-2 virus, and the following two times were when they were injected through immunization. Every contact with the antigen altered the immune system, causing a burst of antibody production lasting up to 14 days [22,23]. The triple exposure of this group caused the most significant rise in neutralizing antibodies among all groups observed.

Although group G5 (*N* = 4), vaccinated people who developed fully symptomatic COVID-19, was excluded from further statistical analysis due to its small size, the results that we obtained for this group correlate with literature data, which show the most significant rise in antibody concentration in vaccinated and infected people [8,21].

### 5.1. Who Needs the Third Dose of the Vaccine?

Synthesis of IgM antibodies is the first phase of the humoral response after initial contact with the antigen. IgG antibody synthesis occurs after the second or repeated contact with the same antigen. Analysis of concentrations of specific IgG and IgM SARS-CoV-2 antibodies and their mutual dependence enabled the identification of early and late phases of the humoral response to SARS-CoV-2 infection and the COVID-19 vaccine [17,21]. Indirectly, it allowed us to identify individuals who had asymptomatic COVID-19. Importantly, the timing of blood sample collection is fundamental for the correct evaluation of the concentrations of IgM and IgG antibodies. This timing depends on the patient’s clinical status and dynamic changes in the immune system, e.g., after SARS-CoV-2 infection, and can effectively optimize the planning of the next vaccine dose against the Delta variant [22]. Such an evaluation must compare clinical examinations, identify infection symptoms or lack thereof, and perform a serological investigation for the presence virus antigens or lack thereof.

In Israel, we observed an increase in SARS infection among people vaccinated with the BNT1262b vaccine 146 days after vaccination [24]. This increase was most pronounced in people over 60 [24]. In our opinion, it is reasonable to start measuring specific IgG (late effect) and IgM (early effect) antibodies for vaccination evaluation at the serological window of neutralizing antibody production. To date, data have shown that the humoral response of neutralizing antibodies after vaccination is rapid and decreases. In addition, in some people who have not been exposed to SARS-CoV-2, late humoral and cellular responses are detected, indicating widespread immunity to SARS in the population [25]. In summary, our results indicate the importance of evaluating and monitoring immunological response parameters in vaccinated healthy people and convalescents, which will help establish the need for third and subsequent doses of the vaccine against SARS-CoV-2. 

### 5.2. Conclusions

Although our results could not be correlated with age, an increase in antibodies after systemic BNT162b2 vaccination, as indicated by other authors, confirms the benefits of monitoring antibody levels due to individual humoral responses and the age-associated decrease in immunity [25]. Studies by Liao et al. [26] indicate that specific SARS-CoV-2 IgG is a more sensitive measure than S-RBD antibodies for monitoring the humoral response in asymptomatic COVID-19 patients. The humoral response of cells after COVID-19 seems to be sufficient for more than three months [27]. However, we noted that subjects in our hospital who had a mild illness were not wholly asymptomatic. We suggested individually checking antibodies strongly correlated with the patient’s clinical condition [28,29]. The research reported by Dan et al. [28] agrees with our observations that S-RBD antibodies are stable for up to 6 months. In conclusion, the determination of SARS-CoV-2 IgG provides more important information three months after suffering from the disease or receiving the vaccine. This manuscript was submitted as a preprint.

## Figures and Tables

**Figure 1 vaccines-10-00334-f001:**
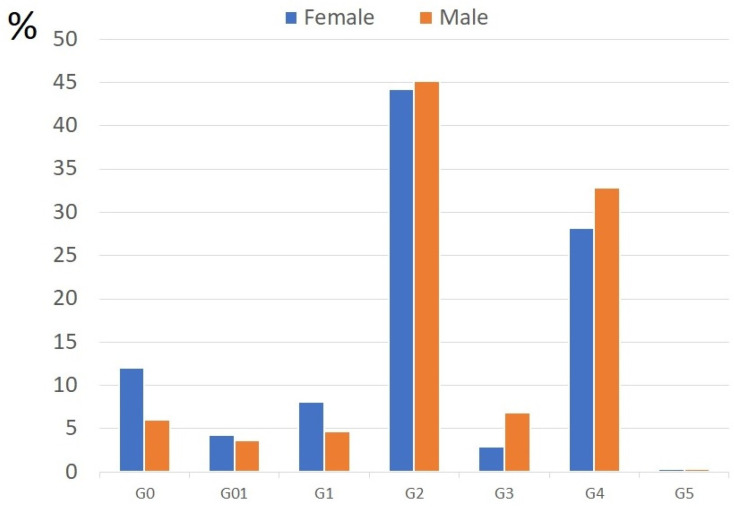
Gender distribution in the study groups. There is no statistically significant difference among the study groups in terms of gender.

**Figure 2 vaccines-10-00334-f002:**
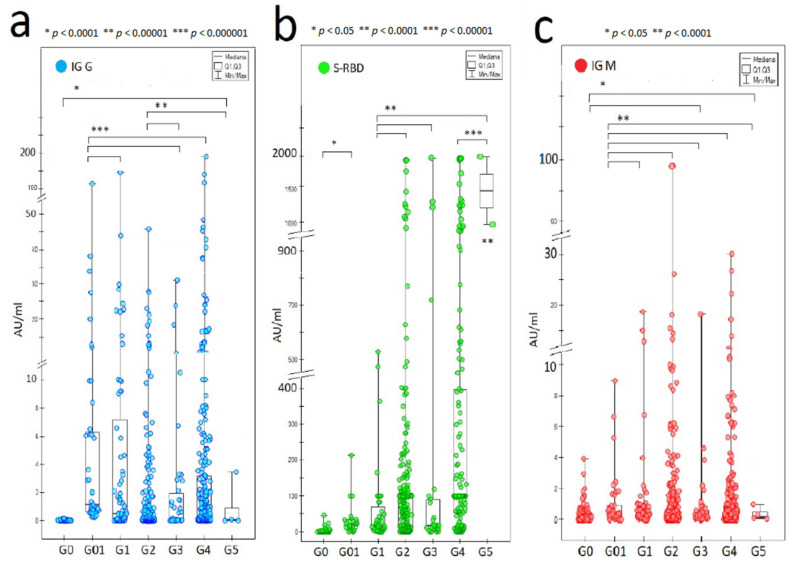
The absolute concentrations of SARS-CoV-2 IgG (**a**), IgM (**c**), and S-RBD IgG (**b**) antibodies in each study group, G0-G5 (*p* < 0.05 appropriately; Kruskal–Wallis ANOVA).

**Figure 3 vaccines-10-00334-f003:**
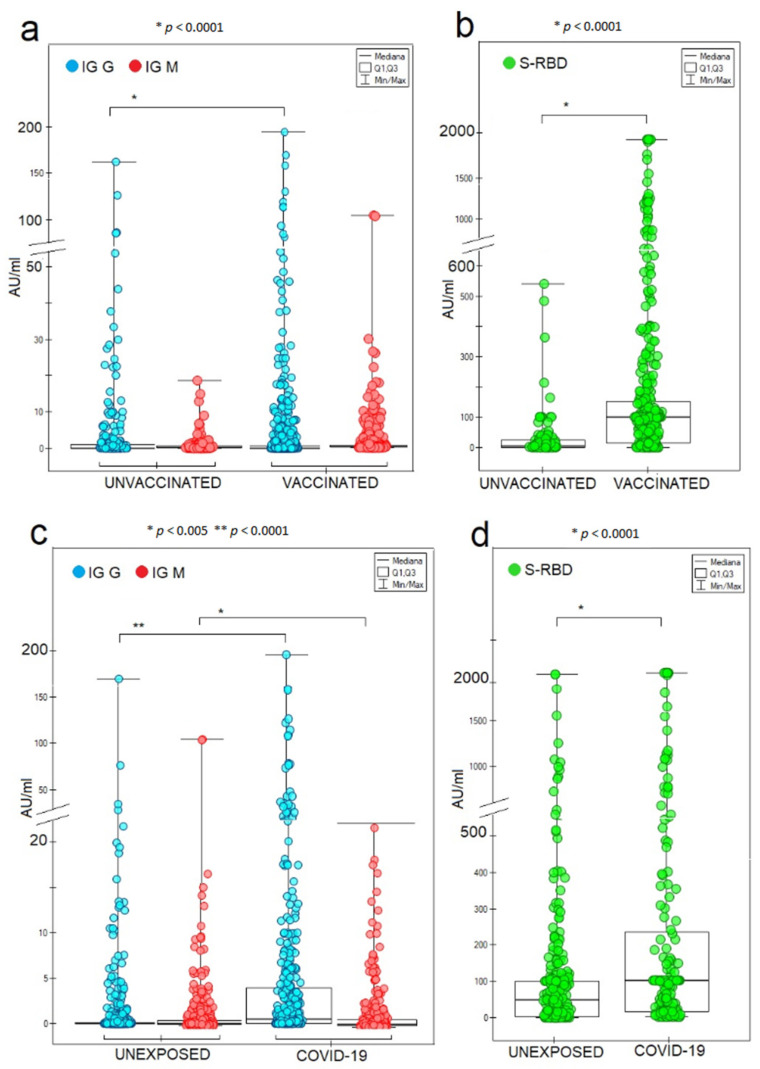
The absolute concentrations of SARS-CoV-2 IgG and IgM antibodies (**a**,**c**) and S-RBD-neutralizing IgG (**b**,**d**) for vaccinated and COVID-19-positive individuals (*p* < 0.005 appropriately; Kruskal–Wallis ANOVA). The unexposed COVID-19 group includes people who tested negative for COVID-19 and people with no history of being infected before measuring the antibodies.

**Figure 4 vaccines-10-00334-f004:**
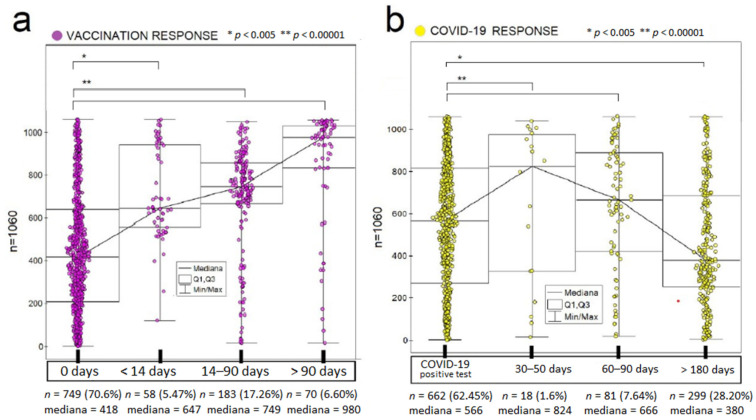
The determination of antibodies in the study groups at intervals (**a**) after full vaccination and (**b**) after obtaining a positive result for SARS-CoV-2 (COVID-19 positive test) (*p* < 0.005 appropriately; Kruskal–Wallis).

**Table 1 vaccines-10-00334-t001:** Population characteristics and outcomes in the case–control study.

Groups	Participants	G0	G01	G1	G2	G3	G4	G5
All	1060	111	43	76	472	42	312	4
%	100	10.69	4.05	7.17	44.52	3.96	29.43	0.38
2020	499 (47.07%)	51	14	30	215	7	179	3
2021	561 (52.92%)	60	29	46	257	35	133	1
**Sex**								
Female	783 (73.87%)	94	10	33	63	346	23	3
Vaccinated	594 (75.86%)	0	0	0	347	23	221	3
Unvaccinated	189 (24.14%)	93	33	63	0	0	0	0
Male	277 (26.13%)	18	33	10	125	19	91	1
Vaccinated	236 (85.19%)	0	0	0	125	19	91	1
Unvaccinated	41 (14.8%)	18	10	13	0	0	0	0
**Age range**								
<35	220 (20.75%)	39	8	20	102	4	47	0
36–49	361 (34.06%)	39	12	23	163	15	109	0
>50	479 (45.18%)	34	23	33	206	23	156	4
**COVID-19 test**								
Negative	671 (63.3%)	111	43	0	471	35	11	0
Positive	389 (36.69%)	0	0	76	1	7	301	4
**COVID-19 response**								
No	662 (62.45%)	111	43	0	470	35	2	0
30–50 days	18 (1.69%)	0	0	13	1	2	1	2
60–90 days	81 (7.64%)	0	0	18	0	3	58	2
>180 days	299 (28.2%)	0	0	45	1	0	251	0
**Vaccinated**								
No	233 (21.79%)	111	43	76	0	0	0	0
Yes	827 (78.01%)	0	0	0	472	24	312	4
Comirnaty	787 (74.2%)	0	0	0	472	0	312	4
AstraZeneca	21 (1.98%)	0	0	0	0	21	0	0
Moderna	19 (1.79%)	0	0	0	0	19	0	0
J&J	2 (0.19%)	0	0	0	0	2	0	0
**Vaccinated response**								
No	749 (70.66%)	111	43	76	288	31	197	2
<14 days	58 (5.47%)	0	0	0	28	4	26	0
14–90 days	183 (17.3%)	0	0	0	111	0	64	1
>90 days	70 (6.6%)	0	0	0	45	0	25	1
**IgG (AU/mL)**	1060							
<1.0	685 (64.62%)	111	19	41	420	27	192	3
>0.2	375 (35.37%)	0	43	42	103	19	167	1
>1.0	247 (23.30%)	0	24	35	52	15	120	1
**IgM (AU/mL)**	1060							
<1.0	879 (82.92%)	103	33	65	397	36	242	3
>1.0	181 (17.07%)	18		11	75	6	70	1
**S-RBD IgG (AU/mL)**	546							
<1.0	81 (7.64%)	31	2	6	31	4	7	0
>1.0	465 (43.86%)	20	22	33	220	21	147	2
>50	295 (27.83%)	0	3	11	154	8	117	2
>100	246 (23.20%)	0	3	8	126	6	101	2
>500	57 (5.38%)	0	0	1	14	4	36	2
>1000	38 (3.58%)	0	0	0	10	3	23	2

The average age of 1060 participants is 47.5 years. The population is normally distributed. Preferential extension of vaccinations to only some age groups in Poland prevented us from correctly interpreting these data. No relationship with gender was observed in the analyzed groups. All antibody concentration results are in AU/mL. The COVID-19 test indicates population groups testing positive or negative by RT-PCR. COVID-19 response indicates the population groups wherein antibodies had a temporal correlation with infection, as measured by a positive test. Vaccinated COVID-19 represents the vaccine population groups. Vaccination response refers to the population groups with a temporal correlation with the performance of the vaccination. We rejected 3 of 1063 participants, regarded as hyper-responders based on IgM and IgG because of markedly higher measured antibody concentrations (>2000 AU/mL IgG, >700, and >300 AU/mL IgM). The person with an over-response of IgG was also an early responder (<14 days) due to an autoimmune disease. The G5 group included too few people to draw conclusions, but four people had high levels of S-RBD IgG and low levels of SARS-CoV-2 IgM and IgG. This phenomenon seems to be related to the loss of antibodies following a previous complete vaccination and an incidence of SARS-CoV-2 infection. This study is most relevant for Delta virus variants in Poland. We were unable to determine the type of virus variants in our laboratory.

## Data Availability

Data are available on request.

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
