# Peer review of "Systemic COVID-19 Vaccination Enhances the Humoral Immune Response after SARS-CoV-2 Infection: A Population Study from a Hospital in Poland Criteria for COVID-19 Reimmunization Are Needed"

_vaccines, 2022, doi:10.3390/vaccines10020334_

Round 1
Reviewer 1 Report
Summary: A study aims to compare the intensity of IgG, IgM, and neutralization Spike IgG in infected and in vaccinated cohorts. The vaccination is by major approved types of vaccines. The results show the higher concentration of antibodies in the group of vaccination after recovery from Covid-19. The authors propose to monitor the IgG antibodies level prior to the next vaccination.
Major comments:
- The title and the abstract are inconsistent. The focus of the study on oncology patients is not mentioned throughout or addressed at all. This remains confusing.
- Add /comment on sex partition for the groups in Table 1 (if not an important feature, it is worth mentioning it). Also, analysis for the (largest) group of vaccinated the age relation to assessing post-vaccination antibody with age.
- No indication of any specific variants so neutralization and policy of antibodies titer is variant-specific. There is no information on this aspect (even before Omicron)
- Key publications on the topic are missing such as on the ‘antibodies typing and sensitivity for asymptomatic doi: 10.3389/fimmu.2021.724763. The time lag from vaccination https://doi.org/10.1016/j.lanepe.2021.100208. Memory over time 10.1126/science.abf4063 (2021), simulation and data for titer of S-antibodies 10.1038/s41467-021-21444-5 and alike.
- As the data points are spread across order of magnitude, it could be that Q1-Q4 will be more appropriate. How will it affect the analysis?
- In discussion: line 158 on hierarchy. Is it resistant to the time scale from the exposure? sex? Age?
- Data are available on request- better to provide a table in the supplemental with data. Often it is a journal policy request).
Minor comments:
- Figures 1 and 2 are the main results of the manuscript. It is not easy to follow due to the used varying scales. Consider showing a cleaner view of only the Q1-Q3 box (or even the mean with SEM in the appropriate scale of AU/ml) as a separate view. Violin plots in a unified scale for each of the tests (log if needed; remove outliers if needed) are often easier to look at. Also, G5 could be removed from Fig 1 and discussed separately (as done in Fig 2).
- In the introduction it is stated The Pfizer-BioTech vaccine …. Change to ‘among the most effective’ (some people will argue that the Moderna vaccine is better).
- Change COMIRNATY to Comintary. Also, add some details on the assay (its sensitivity) or add a reference to it in the Method section.
- Change the wording “line 48: COVID-19 patients at regular times” (not clear, ‘regular times’)
- Explain: line 48 -‘Higher levels’ – relative to what?
- Explain: line 62- ‘Aging of the immune system’ - what do you mean?
- Fig 3b, should it be >30 days? ( or 30-50?) Please clarify (or remove and add to the other groups)
- In discussion, for the group of asymptomatic - is the identification of such group meets the prevalence of asymptomatic?
- Line 163: Vaccinated people never met the virus but instead the spike protein of the virus. Please correct
- Correct lines 185, 192 to SARS-CoV-2 infection
- The manuscript was sent as a preprint with Ref 26 that is missing.
Reviewer 2 Report
Estimated Authors of the paper "Systemic COVID-19 vaccination also enhances the humoral immune response after SARS CoV-2 infection in the population of an oncology hospital in Poland. Criteria for COVID-19 re-immunization are needed"
I've read your original contribution with great expectatives and interest. While SARS-CoV-2 pandemic is still ongoing, every piece of information on the efficacy of vaccines and EBM for recommending repeated shots are urgently in need. ccording to your text, you did perform an observation study on 1063 individuals, including various subgroup: vaccinated with Comirnaty, vaccinated with AZ vaccine, with J&J formulate, individuals with natural immunity following infection - the latter not vaccinated, vaccinated with adenoviral or mRNA-based formulates. Unfortunately, I'm committed to recommend the eventual rejection of this paper for the following reasons.
- I had some significant difficulties in grasping the very meaning of your research. Serological estimates are reported in a quite confusing way. Table 1, for examples, is redundant in terms of reporting estimates for immunoglobulines, both as categorical values, as continuous values, including data on IgM (whose significance when dealing with immunity is potentially even more conflicting than that associated with IgG). Moreover, the variable "Vaccination response" is not clearly stated.
- The groups are of strikingly different size. As a consequence, statistical analyses are of doubtful significance when dealing with vaccines other than COMIRNATY. I warmly recommend to adjust your analyses by taking in account only not-vaccinated vs. comirnaty.
- Characteristics of the study groups are lacking and must be reported. We cannot rule out, according to your report, that some of the differences you identified (but, see point 2) were associated with basal characteristics of the patients rather than with specificities of the formulates.
Several minor remarks:
- English text must benefit from a professional English editor or an extensive editing;
- Some statement across the text are inappropriate. For instance, even though the efficacy of COMIRNATY is not in discussion, stating that it represents the "most effective vaccine" is not correct, as MODERNA has nearly the very same efficacy profile;
- Discussion and conclusions would benefit from their dichotomization, the latter with the inclusion of some keypoints or more detailed "take home message" as a closing statement.
Author Response
Vaccines -1544304
Thank you note
Thank you for improving our work, and we assure you that we have made every effort to improve the work and take into account all the detailed amendments that we were able to change to the final version of the manuscript.
Authors
Major remarks
Reviewer 2
- I had some significant difficulties in grasping the very meaning of your research. Serological estimates are reported in a quite confusing way. Table 1, for examples, is redundant in terms of reporting estimates for immunoglobulines, both as categorical values, as continuous values, including data on IgM (whose significance when dealing with immunity is potentially even more conflicting than that associated with IgG). Moreover, the variable "Vaccination response" is not clearly stated.
Our studies were conducted in an oncology hospital with volunteers taking antibody measurements, usually before and after vaccination. Qualification for measurements was made based on available advice, and it is estimated that over 60% were former patients. Approximately 40% are medical and non-medical personnel who also had access to research. Retrospectively, we searched for people with humoral immunity after infection with SARS CoV-2, and when vaccines appeared, we observed different humoral responses of IgM and IgG levels, trying to respond to which asymptomatic patients have elevated antibodies. Therefore, Table 1 was created as a supplement.
We have introduced the correction for Reviewer 1
We suggest a caption beneath Table 1.
The average age of 1060 participants is 47.5 years. There is normal population distribution. All antibodies concentration results are in AU/ml. The COVID-19 test represents population groups tested positive or negative RT-PCR. COVID-19 response represents the population groups where antibodies with a temporal correlation with infection tested, as measured by a positive test. Vaccinated COVID-19 represents the vaccine population groups. Vaccination response refers to the population groups with a temporal correlation with the performance of the vaccination. We rejected from 1063 participants three persons assumed as hyper responders IgM and IgG because of spectacular higher antibodies concentrations measured (>2000 AU/ml IgG and >700 and >300 AU/ml IgM). The person with an over-response to IgG was also an early responder (<14 days) due to an autoimmune disease. In the G5 group too few people to conclude, but four people had high levels of S-RBD IgG and low levels of SARS CoV-2 IgM and IgG. This phenomenon seems to be related to the loss of antibodies following a previous complete vaccination and the duplication of SARS CoV-2 infection.
Table 2 is data on gender and age supplement to Table 1.
|
Groups |
Participants |
G0 |
G01 |
G1 |
G2 |
G3 |
G4 |
G5 |
|
All |
1060 |
111 |
43 |
76 |
472 |
42 |
312 |
4 |
|
% |
100 |
14.62 |
4.05 |
7.17 |
45.66 |
3.96 |
29.43 |
0.38 |
|
Sex |
|
|
|
|
|
|
|
|
|
Female |
783(73.87%) |
94 |
10 |
33 |
63 |
346 |
23 |
3 |
|
Male |
277(26.13%) |
18 |
33 |
10 |
125 |
19 |
91 |
1 |
|
Age range |
|
|
|
|
|
|
|
|
|
< 35 |
220(20.75%) |
39 |
8 |
20 |
102 |
4 |
47 |
0 |
|
36-49 |
361(34.06%) |
39 |
12 |
23 |
163 |
15 |
109 |
0 |
|
>50 |
479(45.18%) |
34 |
23 |
33 |
206 |
23 |
156 |
4 |
Extending preferential vaccinations to only some age groups in Poland prevented us from correctly interpreting these data. No relationship with gender is in the analyzed groups.
- The groups are of strikingly different size. As a consequence, statistical analyses are of doubtful significance when dealing with vaccines other than COMIRNATY. I warmly recommend to adjust your analyses by taking in account only not-vaccinated vs. comirnaty.
We agree with the suggestions. In future work, we will follow the recommended tips. We plan to cover the entire year of 2021, when field vaccinations were carried out in hospitals for cancer patients and employees. We have over 4,000 samples.
- Characteristics of the study groups are lacking and must be reported. We cannot rule out, according to your report, that some of the differences you identified (but, see point 2) were associated with basal characteristics of the patients rather than with specificities of the formulates.
We performed the characteristics of the research groups retrospectively on the basis of medical history and vaccination. We did not select patients into groups based on their illness, duration of treatment, or recovery from illness. We did not plan such a selection. In our future work, we will select clinical groups from 4,000 subjects.
Several minor remarks:
- English text must benefit from a professional English editor or an extensive editing;
We will follow the suggested recommendations. We use Grammarly premium.
- Some statement across the text are inappropriate. For instance, even though the efficacy of COMIRNATY is not in discussion, stating that it represents the "most effective vaccine" is not correct, as MODERNA has nearly the very same efficacy profile;
We will follow the suggested recommendations.
The Pfizer-BioNTech vaccine BNT162b2 (Comirnaty) [1] is recognized so far as among the most effective vaccine in preventing SARS Co-V-2 infection and severe COVID-19 [2,3,4].
- Discussion and conclusions would benefit from their dichotomization, the latter with the inclusion of some keypoints or more detailed "take home message" as a closing statement.
The results initially indicate that we should monitor the concentration of SARS COV-2 IgG. According to Prof. Anna Stasiak-Barmuta, we cannot make more precise conclusions because, as we mentioned, vaccinations were planned for specific population groups in Poland. All age groups will be taken into account in our subsequent work. Our suggestions indicate that it is worth measuring the level of antibodies before the next dose of the vaccine.

Round 2
Reviewer 1 Report
The authors address several of the points raised but their response is partial and not always fully satisfactory.
I encourage them to improve the manuscript a bit more, prior to publication. I mention only a few minor issues that I believe are important to address. I realize that the team have hard time to make substantial changes and expect to incorporate some of the suggestions in future reports.
- I still find the title extremely long and inconsistent. It can be read as if the population are oncology patients A possible alternative “Immunization of people that recovered from COVID-19 yielded an enhanced humoral immune response” the addition of Oncology hospital is confusing and could be misleading. If needed one can add - A population study from an hospital in Poland
- Table 2 provides the missing information on sex. It would be useful to do such a partition. Even coloring the people in the graphs by sex could be useful. (if the statistics is too weak, please mention this as a reason not to do the analysis by gender).
- Mention that this study is probably relevant to the Delta variants (add a clear time line). It is OK to mention that you cannot account for the types of variants.
- I understand that the COVID-19 publications are very dynamic. It is quite easy to add a few relevant publications that provide a further support for your main findings (i.e. enhanced response in IgG for COVID-19 recovered people.
Reviewer 2 Report
Estimated Authors of the paper
"Systemic COVID-19 vaccination also enhances the humoral immune response after SARS CoV-2 infection in the population of an oncology hospital in Poland. Criteria for COVID-19 re-immunization are needed"
I've read with interest your revised version.
Unfortunately, Authors have not substantially replied to my previous concerns regarding methodological issues - but none of them has been addressed.
For example: regarding sample size, Authors write: "
We agree with the suggestions. In future work, we will follow the recommended tips. We plan to cover the entire year of 2021, when field vaccinations were carried out in hospitals for cancer patients and employees. We have over 4,000 samples." I'm confident you will be able to publish your report in a highly impacted journal, but we're dealing with the present study and its limits. Therefore this issue remains unsolved.
Similarly, the point about the sample characteristics remains unsolved ("...
We performed the characteristics of the research groups retrospectively on the basis of medical history and vaccination. We did not select patients into groups based on their illness, duration of treatment, or recovery from illness. We did not plan such a selection. In our future work, we will select clinical groups from 4,000 subjects.").
As the key points of the previous review remain not addressed, I'm forced to recommend the rejection of this study in its current status.
Round 3
Reviewer 2 Report
After previous reviews, Authors have improved the overall quality of this paper, addressing as much as possible my previous concerns.
In fact, I think that the paper my accepted in its present status.